# COVID-19 and Bell's Palsy

Eve N. Tranchito [1,2], Amanda Goslawski [1,2], Claudia I. Cabrera [1,2], Cyrus C. Rabbani [1,2], Nicole M. Fowler [1,2], Shawn Li [1,2], Jason E. Thuener [1,2], Pierre Lavertu [1,2], Rod P. Rezaee [1,2], Theodoros N. Teknos [1,2] and Akina Tamaki [1,2,*]

1 Department of Otolaryngology-Head and Neck Surgery, University Hospitals Cleveland Medical Center, Cleveland, OH 44106, USA
2 School of Medicine, Case Western Reserve University, Cleveland, OH 44106, USA
* Correspondence: akina.tamaki@uhhospitals.org; Tel.: +1-216-844-7547

**Definition:** There are various neurological manifestations of coronavirus disease 2019 (COVID-19). Recent data suggest a connection between hemifacial paralysis, or Bell's palsy, and COVID-19. Although the etiology of Bell's palsy is unknown, the leading proposed etiology is viral in nature. Since the onset of the pandemic, numerous studies have investigated the relationship between Bell's palsy, COVID-19 infection, and COVID-19 vaccination. The researchers studied the current literature on the topic of COVID-19 as it relates to Bell's palsy.

**Keywords:** coronavirus disease 2019; Bell's palsy; facial paralysis; vaccination





## 1. Introduction

Bell's palsy (BP) is defined as idiopathic unilateral paresis or paralysis of cranial nerve VII, the facial nerve. It clinically manifests as an acute onset weakness of the muscles that control facial expression. Patients may notice drooling from the corner of the mouth, inability to completely close the eye, or facial asymmetry. Less commonly, patients may develop facial pain, numbness, alterations in taste, or sensitivity to sound. It occurs as hemi-facial paralysis (on one side) and may be partial or complete in regard to facial nerve dysfunction. It affects about 15 to 30 out of 100,000 people per year with a peak incidence between age 15 and 40 [1]. The condition equally affects men and women, however pregnant women may be at a slightly elevated risk [2]. Although the cause of BP is yet to be determined, the leading proposed etiologies are viral, ischemic, and/or immune mediated. One proposed mechanism is re-activation of the herpes simplex virus at the geniculate ganglion of the facial nerve.

Although inconsistent, studies have demonstrated a high incidence of the herpes simplex virus in cases of BP [3]. About 70% of people will fully recover facial nerve function; however, the remaining 30% are left with varying degrees of dysfunction [2]. BP is a diagnosis of exclusion, and it is important to consider other causes of facial nerve weakness, such as parotid lesions or skull base tumors, congenital or syndromic conditions, trauma, and known associated infections, such as Lyme disease or herpes zoster.

Clinical improvement occurs within three weeks of onset in 85% of people and within three to five months in the remaining 15% [2]. The House–Brackmann scale is a numeric point system typically used to measure the degree of paresis and grades recovery; it is based on grading facial symmetry and movement. Treatment at the onset of BP includes systemic corticosteroids and antivirals. Depending on the degree of dysfunction, eye precautions may be recommended to prevent exposure keratopathy in cases of incomplete eye closure. Given the overall good prognosis within 3 months, incomplete recovery should prompt workup for alternative causes of the paralysis, including magnetic resonance imaging of the brain. Physical rehabilitation, chemodenervation, and even surgical procedures may be employed in cases of incomplete recovery.

Coronavirus disease 2019 (COVID-19) is caused by the novel virus, severe acute respiratory syndrome coronavirus-2 (SARS-CoV-2). Its clinical symptomatology is most commonly characterized by respiratory illness; however, serious complications and a wide array of less predictable side effects have challenged patients, clinicians, and scientists. Acute respiratory distress syndrome, acute heart and kidney injury, disseminated intravascular coagulation, and sepsis are some of the more consequential complications of COVID-19. Since the onset of the pandemic, there have been numerous reported neurologic manifestations of the illness; anosmia (loss of smell) and ageusia (loss of taste) were two very common symptoms of COVID-19 early on. In a systematic review of 37 articles released prior to May 20th, 2020, the most common neurologic manifestations of the disease were myalgia, headache, altered sensorium, hyposmia, and hypogeusia. Other manifestations of COVID-19 included myelitis, Guillain–Barre syndrome, encephalopathy, ischemic stroke, and intracerebral hemorrhage [4].

Another neurologic manifestation of COVID-19 with numerous reports in the literature is BP. In addition to COVID-19 infections, the association of the COVID-19 vaccine and BP has also been controversial. This manuscript explores the available literature on this topic, none of which provide definitive evidence of the presence, or lack of, an association of BP with COVID-19 infection or vaccination.

## 2. Bell's Palsy and Association with COVID-19 Infection

The emerging literature suggests that BP is a potential neurologic sequelae of COVID-19. There are numerous case reports published describing patients who presented with BP and were found to test positive for COVID-19 (Table 1). Iacono et al. reported a case of one child presenting with BP who tested negative for COVID-19 on a nasopharyngeal swab, but positive on serology [5]. Another patient who presented with facial paralysis had signs of COVID-19 on chest imaging and was eventually found to have positive COVID-19 serology [6]. In a case series of eight patients with BP and COVID-19, three had BP as the presenting symptom [7]. These examples raised concern that COVID-19 may cause BP, thus prompting more robust studies.

In a systematic review of the literature published between March 2020 and December 2021, there were 32 relevant publications with 46 total patients who had BP as the only major neurologic manifestation of COVID-19. In 37% of these cases, BP was the initial symptom and in 63% it developed after the onset of other COVID symptoms [8]. One prospective cross-sectional study tested all patients with BP for SARS-CoV-2 IgM and IgG. Of 41 subjects, 24.3% had positive antibodies. This is a higher seroprevalence than that seen in similar studies of asymptomatic patients, suggesting COVID-19 infection may be a risk factor for BP [9]. Another study suggested the same when they examined 38 cases of BP between February and May 2020 and found that eight patients had active or recent COVID-19 symptoms compared to only 2 of 22 the year before [10]. Furthermore, in a population-based cohort and self-controlled case series out of Spain and the United Kingdom, rates of BP were higher than expected with a standardized incidence ratio of 1.33 [11]. Tamaki et al. utilized a multi-institutional database of 41 healthcare organizations and found a higher number of patients than expected who were diagnosed with BP within eight weeks of COVID-19 infection. The study suggested a relative risk of 6.8 (95% CI: 3.5–12.2 $p < 0.001$) of BP in those diagnosed with COVID-19 [12].

The current literature on this topic is conflicting, with some reports suggesting that there is no association between BP and COVID-19 infection. Mutlu et al. examined all patients who presented to their hospital due to facial paralysis and found that only 2 of 153 had a COVID-19 diagnosis [13]. Another study in Spain studied the incidence of BP and COVID-19 during different pandemic waves by comparing background incidence to that of different "peaks" of the pandemic. They found that cases of BP did not change during times of higher COVID-19 infections, suggesting BP is not necessarily influenced by COVID-19 infection [14].

**Table 1.** Literature on the association between Bell's palsy and COVID-19 infection.

| Title | Author | Year | Study Design | Major Findings/Conclusions |
|---|---|---|---|---|
| Surge of Bell's Palsy in the era of COVID-19: Systematic review [8]. | Gupta et al. | 2022 | Systematic Review | • BP can be found as the only neurological manifestation of COVID-19<br>• Suggests COVID-19 as another viral etiology of BP |
| Facial paralysis as the only symptom of COVID-19: A prospective study [9]. | Islamoglu et al. | 2021 | Prospective longitudinal study. | • Patients presenting with BP have a higher seropositivity rate of COVID-19 than asymptomatic patients<br>• Facial paralysis could be the only symptom of COVID-19 infection |
| Facial palsy during the COVID-19 pandemic [10]. | Codeluppi et al. | 2021 | Retrospective case-control | • Higher rate of BP during COVID-19 pandemic than year prior |
| Association between covid-19 vaccination, SARS-CoV-2 infection, and risk of immune mediated neurological events: population based cohort and self-controlled case series analysis [11]. | Li et al. | 2022 | Population based cohort and self-controlled case series analysis | • Higher than expected incidence of BP after COVID-19 infection<br>• Standardized incidence ratio for BP was 1.33 (95% CI: 1.02–1.74) |
| Incidence of Bell Palsy in Patients with COVID-19 [12]. | Tamaki et al. | 2021 | Retrospective longitudinal cohort study | • Higher number of patients than expected diagnosed with BP within 8 weeks of COVID-19 infection<br>• Relative risk of 6.8 (95% CI: 3.5–12.2, $p < 0.001$) of BP in those diagnosed with COVID-19 |
| Does the SARS-CoV-2 pandemic really increase the frequency of peripheral facial palsy [13]? | Mutlu et al. | 2021 | Retrospective cohort | • Number of patients with BP during COVID-19 pandemic was similar to previous years |
| Data from 235 Cases of Bell's Palsy during COVID-19. Pandemic: Were There Clusters of Facial Palsy [14]? | Martin-Villares et al. | 2021 | Retrospective cohort | • BP incidence did not differ from background incidence throughout 4 "peaks" of COVID-19 |

## 3. Bell's Palsy and Association with COVID-19 Vaccination

In addition to COVID-19 infections, the literature has also examined whether BP can be caused by the COVID-19 vaccine. Vaccines for COVID-19 were produced at an unprecedented rate; a process that typically took 8 to 15 years was achieved in less than one full year. Additionally, the use of an mRNA vaccine was the first of its kind approved for human commercial use. Multiple factors, including these, contributed to public hesitancy surrounding the COVID-19 vaccine. The three most commonly studied COVID-19 vaccines within this entry were BNT162b2, an mRNA vaccine, the ChadOx1 nCoV-19, a viral vector vaccine, and the mRNA-1273-nCoV, another mRNA vaccine. Anecdotal reports of adverse events associated with the vaccine were followed by published reports, further instilling hesitancy in those considering vaccination against COVID-19 (Table 2).

**Table 2.** Literature on the association between Bell's palsy and COVID-19 vaccination.

| Title | Author | Year | Study Design | Major Findings/Conclusions |
|---|---|---|---|---|
| Safety of COVID-19 vaccination and acute neurological events: A self-controlled case series in England using the OpenSAFELY platform [15]. | Walker et al. | 2022 | Self-controlled case series | • Increased risk of BP after ChAdOx1 (RR: 1.39, 95% CI: 1.27–1.53)<br>• 17.9 more cases of BP per 1 million vaccines<br>• Absolute risk is low (<1/60,000) |
| Adverse events of special interest and mortality following vaccination with mRNA (BNT162b2) and inactivated (CoronaVac) SARS-CoV-2 vaccines in Hong Kong: A retrospective study [16]. | Wong et al. | 2022 | Retrospective population-based cohort | • Higher rate of BP following the first dose of CoronaVac vs BNT162b2 (67 versus 30 per 100,000 person-years; RR: 1.95, 95% CI: 1.12–3.41, *p-value*: 0.018) |
| Association between vaccination with the BNT162b2 mRNA COVID-19 vaccine and Bell's palsy: a population-based study [17]. | Shibli et al. | 2021 | Retrospective longitudinal | • Possible association between BNT162b2 mRNA vaccine and BP<br>• Standardized incidence ratios of 1.36 (95% CI: 1.14–1.61) and 1.16 (95% CI: 0.99–1.36) for first and second dose, respectively |
| Association of COVID-19 Vaccination and Facial Nerve Palsy: A Case-Control Study [18]. | Shemer et al. | 2021 | Retrospective matched case-control | • No association between BP and BNT162b2 |
| Hospital-based observational study of neurological disorders in patients recently vaccinated with COVID-19 mRNA vaccines [19]. | Koh et al. | 2021 | Prospective observational cohort | • No association between BP and three mRNA vaccines, BNT162b2 and mRNA-1273 |
| Risk of serious adverse events after the BNT162b2, CoronaVac, and ChAdOx1 vaccines in Malaysia: A self-controlled case series study [20]. | Ab Rahman et al. | 2022 | Self-controlled case series | • No increased risk of BP after BNT162b2, CoronaVac, and ChAdOx1 vaccines in Malaysia |

Numerous case reports of BP following vaccine administration have been published [21–27]. In one, a 60-year-old man with Human Immunodeficiency Virus developed facial droop 42 h after receiving Pfizer/BioNTech BNT162b2 [28]. Another patient who interestingly had a history of BP developed it again after the second dose of the Pfizer vaccine [29]. In another report, a 61-year-old man developed BP within five hours of the first vaccine and then again on the contralateral side after the second dose [30]. In a PubMed and Google Scholar review of case reports and case series examining neurological manifestations of COVID-19 vaccination, Sriwastava et al. found that after Guillain–Barre syndrome, BP was the second most common peripheral nervous system manifestation [31]. These reports prompted more robust studies investigating the relationship between the COVID-19 vaccines and potential neurological side effects.

Studies then emerged calculating background incidence in order to determine whether a true change in BP incidence exists in relation to COVID-19 vaccinations. Researchers in Ontario, Canada, conducted a population-based retrospective observational study to estimate the background incidence of BP, finding the pre-pandemic mean annual rate of BP was 2.8 per 100,000 from the years 2015 to 2019 [32]. A multinational network cohort study including databases from Australia, France, Germany, Japan, Netherlands, Spain, United Kingdom, and United States, with a sample size of 126,661,070, notably found large variations between databases of different countries, suggesting rates of adverse events including BP should be compared to more localized background incidences [33].

Some studies claim an increased risk of BP in vaccinated patients. In a self-controlled case series in England, authors found a significantly increased rate of BP (17.9 cases per one million) following ChAdox1 vaccines; however, no association was found with the mRNA-1273 and BNT162b2 vaccine [15]. In another study comparing the inactivated CoronaVac vaccine and the mRNA-based vaccine, BNT162b2, authors conducted a retrospective study, based in Hong Kong, comparing the incidence of adverse events and all-cause mortality between the two. They analyzed a cohort of people who received at least one dose of either vaccine and found that among over 2 million people, there was a significantly higher rate of BP following the first dose of CoronaVac vs. BNT162b2 (67 versus 30 per 100,000 person-years; IRR = 1.95, 95% CI: 1.12–3.41, *p* = 0.018) [16]. In a study using a database of the largest healthcare system in Israel, there were 132 cases of BP out of 2,594,990 patients vaccinated with BNT162b2 after the first dose and 152 cases out of 2,434,674 s dose of the vaccine. This equated to standardized incidence ratios of 1.36 (95% CI: 1.14–1.61) and 1.16 (95% CI: 0.99–1.36), respectively, suggesting the possibility of an association between BP and the BNT162b2 mRNA vaccine [17]. However, other similarly designed studies failed to find an association between the two. A separate study taking place in Israel, for example, utilized a case-control design to evaluate patients recently vaccinated with BNT162b2. A proportion of patients with BP with exposure to the vaccine were compared to the overall number of patients with facial nerve palsy in prior years, and no association was found [18]. In addition, in a prospective study involving seven hospitals in Singapore, only 11 of 1,398,074 patients who were vaccinated with BNT162b2 and mRNA-1273 developed BP, and the authors were not able to establish any association [19]. In a self-controlled case series study assessing the risk of serious adverse events after BNT162b2, CoronaVac, and ChAdOx1 vaccines in Malaysia, authors were unable to find any increased risk of BP following either dose of all three vaccines 21 days after vaccination [20].

In two phase 3 COVID-19 vaccine trials, there were eight reported cases of BP amongst 73,868 total participants. Seven of these cases occurred in the vaccine group, which equates to an incidence of 19 per 100,000 [12,34,35]. The US Food and Drug Administration and the UK Medicine and Healthcare Products Regulatory Agency argue that these numbers are similar to the background rates of BP and in a disproportionality analysis by Renoud et al., there was no difference in reported adverse events of BP after COVID-19 vaccines compared to other viral vaccines [36]. It is important to note that facial paralysis is a known adverse event of vaccines, and that this phenomenon is not isolated to the COVID-19 vaccines; in a review of the vaccine adverse event reporting system (VAERS) database, the influenza vaccine had the greatest number of reported BP events, followed by varicella, and then human papilloma virus. The authors of this study note that the likelihood of BP remains very low overall at 0.26% over a 10-year span [37]. A different study using the VAERS database showed that reported levels of BP following COVID-19 mRNA vaccines are comparable to those of influenza before the COVID-19 pandemic [38].

Authors who found an association between the vaccine and BP conclude that despite this, the absolute or attributable risk is very low, and the benefits far outweigh the risks [17,39–41]. They highlight that the post-vaccine rates of adverse events, including BP, were similar to the expected background rates [11]. Furthermore, Tamaki et al. concluded that those with COVID-19 infection have a higher risk of BP than those who are vaccinated [12].

## 4. Mechanism of COVID-19 Infection and Vaccination and Bell's Palsy

Although there are a number of studies aiming to identify a connection between COVID-19 infection, vaccines, and BP, there is much to be discovered about the potential mechanism to explain such. The leading proposed etiology of BP is viral in nature, suggesting reactivation of HSV-1 at the geniculate ganglion as the possible cause [3,42]. Mouse models have shown the ability to induce facial nerve paralysis by infecting mice with HSV-1 [43]. In a mouse model of HSV-1-induced facial paralysis, Hato et al. found an increased level of nitric oxide, a free radical, and histologic evidence of demyelination

in the mice experiencing facial paralysis. Their study suggested toxicity of nitric oxide on Schwann cells of the peripheral nervous system leading to nerve dysfunction. Their findings were supported by a significantly lower incidence of facial nerve dysfunction seen in mice treated with Edaravone, a free radical scavenger [44]. In the case of Ramsay Hunt Syndrome, facial paralysis is caused by the reactivation of varicella zoster virus, also at the geniculate ganglion [45]. The evidence of viral-induced facial paralysis suggests a possible route for COVID-19 infection to do the same. It has furthermore been proposed that coronaviruses are able to use olfactory nerves as the initial entrance into the central nervous system (CNS). Another hypothesis suggests "neuronal dissemination", a process of active transport that allows access to the CNS [46]. Another proposed mechanism of BP which could explain how COVID-19 causes neuropathy involves the principle of molecular mimicry, during which a foreign antigen, similar in structure to a self-antigen, causes immune-system sensitization. In the case of BP, this is thought to specifically affect the myelin sheath, impairing nerve conduction [47]. Coronaviruses use a glycoprotein that allows viral entry through the angiotensin-converting enzyme 2 receptor, which is often found on glial cells of the central and peripheral nervous system. It is suggested that this mechanism, also termed neurotropism, is one way the virus gains access to the nervous system [48].

It has been postulated that COVID-19 induces a hypercoagulable state, which may explain another way the virus affects our nervous system. In a systematic review by Parra-Medina et al., 60% of COVID-19 autopsies found microthrobi in the lung, heart, kidney, or liver. Furthermore, in a study by Karimi-Galougahi et al., positron emission tomography and computed tomography (PET/CT) obtained on a COVID-19-positive patient who presented with unilateral facial paralysis showed decreased metabolic activity within the affected facial nerve. This finding suggested a proposed mechanism of microthrombus-mediated pathology within the facial nerve or direct SARS-CoV-2 infection of the neurons or glial cells [49].

One mechanism proposing how BP may be caused by the COVID-19 vaccine is based on a theory that the pathogenesis of mRNA-vaccine-related BP is mediated by type I interferons. mRNA vaccines are known to cause a profound type I interferon response, and BP is a reported side effect seen in patients undergoing type I interferon therapy. Furthermore, type I interferon release is known to cause transient lymphopenia, which was the most common hematologic change seen during phase 1 trials of mRNA COVID-19 vaccines [47].

## 5. Conclusions

There is much to be discovered around the topic of COVID-19 as it relates to BP. Case reports of BP following both the COVID-19 vaccine and COVID-19 infection lack substantial evidence but opened the door for more robust studies.

Despite this, there has been no definitive evidence of an association between COVID-19 infection, COVID-19 vaccines, and BP.

Studies have mostly failed to identify incidences of BP that deviate significantly from pre-pandemic means.

The risk of BP following COVID-19 vaccines is no different from that seen with other viral vaccines.

Overall, the benefit of vaccination outweighs the risk of BP and there is insufficient evidence at this time to refute this.

**Author Contributions:** Conceptualization, E.N.T. and A.T.; methodology, E.N.T. and A.T.; formal analysis, E.N.T. and A.T.; investigation, E.N.T. and A.T.; resources, E.N.T. and A.T.; data curation, E.N.T., C.I.C. and A.T.; writing—original draft preparation, E.N.T. and A.T.; writing—review and editing, E.N.T., A.G., C.I.C., N.M.F., C.C.R., T.N.T., J.E.T., P.L., R.P.R., S.L. and A.T.; supervision, A.T. All authors have read and agreed to the published version of the manuscript.

**Funding:** This research received no external funding.

**Conflicts of Interest:** The authors declare no conflict of interest.

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
