# Peer review of "COVID-19 and Bell’s Palsy"

_encyclopedia, doi:10.3390/encyclopedia2040133_

Round 1
Reviewer 1 Report
The article represents an interesting trial to map the association of Bell’s palsy and COVID 19 infection.
Systemic review should be mentioned in Heading.
The method of literature selection should be precisely described. Define the inclusion and exclusion criteria.
I do not like the division to two parts: association with Infection and with vaccination. I would recommend choosing one topic only focused on one problem, make statistical analysis.
COVID 19 is neurotropic virus, however Bell’s palsy is idiopathic, once the patient is COVID 19 positive, the palsy is not idiopathic anymore. The various number of nerves can be affected in SARS COVID 19.
There is mixture of Bell’s Palsy and Facial nerve palsy of known aetiology.
I recommend not use Term Bels Palsy, but Facial nerve paralysis in COVID 19 positivity.
Conclusion is not adequate; it should give more clear statements.
In conclusion, I like the topic, but I would recommend major changes of the article, namely in data processing.
Author Response
Thank you for the reviewer comments. We have made edits to the manuscript to address the reviewer's comments. Please see itamized comments below: The authors must be far more clear about the contradictory evidence available in the literature. I also think they need to be more circumspect when discussing cause/effect or association between SARS-CoV-2, vaccination and Bell's Palsy.- we agree that the current literature is not conclusive on this issue. In order to provide more clarity, we have include the following paragraph in the introductions. Another neurologic manifestation of COVID-19 with numerous reports in the literature is BP. In addition to COVID-19 infections, the association of the COVID-19 vaccine and BP has also been controversial. This manuscript explores the available literature on this topic, none of which provide definitive evidence of the presence, or lack of, an association of BP with COVID-19 infection or vaccination. There are other, more authoritative and comprehensive reviews and discussions not considered by the authors. They must be included (e.g. DOI:https://doi.org/10.1016/S1473-3099(21)00467-9). My reading of the literature tells me that the risk of developing neurological conditions from COVID vaccination is extremely low, but that getting the virus itself raises the risk.- we looked into the cited article "The association between COVID-19 vaccination and Bell's palsy". This is a well done commentary in the Lancet, but this is not an original article and is, therefor, not included in the study. It cites many of the original studies listed in our review and the authors conclusion is very similar to our own. For reference, Cirillo et als conclusion is below: "From a clinical, patient-oriented perspective, none of the studies published so far provide definitive evidence to inform the choice of a specific vaccine in individuals worldwide with a history of Bell's palsy. However, the data published by Wan and colleagues do offer valuable information for a rational and informed choice of COVID-19 vaccines for patients in Hong Kong, and for those in countries where both BNT162b2 and CoronaVac are available. While waiting for conclusive evidence on vaccine-associated facial paralysis, one certainty remains: the benefit of getting vaccinated outweighs any possible risk." Note, this commentary only discussed vaccinations and not infections.
The conclusions need to be far more detailed and presented as bullet points.- The conclusions have been put in bullet form. We have provided some more detail, but it is not possible to make firm or definitive conclusions on this issue as it is still under study and there is no definitive conclusion Please let me know if you have any additional thoughts or edits that we can provide.
Reviewer 2 Report
A concise overview of the papers summarizing the reports of possible association of idiopathic peripheral facial nerve paralysis ("Bell's palsy, BP") with Covid-19 and Covid-19 vaccinations, given the overall incidence of BP pre-covid. Sufficient detail is given to weigh the strengths and weaknesses of the different reports. The conclusions of the overview are appropriate:
1) there appears no convincing evidence of an association with the vaccine; 2) that a virus could cause BP is logical given that other viral infections are thought to be associated with BP. However the reports of covid-19 associations are mixed; if there is an association it appears to be small.
Author Response
A concise overview of the papers summarizing the reports of possible association of idiopathic peripheral facial nerve paralysis ("Bell's palsy, BP") with Covid-19 and Covid-19 vaccinations, given the overall incidence of BP pre-covid. Sufficient detail is given to weigh the strengths and weaknesses of the different reports. The conclusions of the overview are appropriate:
1) there appears no convincing evidence of an association with the vaccine; 2) that a virus could cause BP is logical given that other viral infections are thought to be associated with BP. However the reports of covid-19 associations are mixed; if there is an association it appears to be small.
Reply: Thank you for the review and comments. We agree with this reviewers interpretation of your points in the article.
Reviewer 3 Report
The paper presents current data on the connection between Bell's palsy and Covid-19. This thorough research on the neurological manifestation of Covid-19, based on up-to-date literature, meets the requirements of the Journal and adds information to the general knowledge of Covid-19.
Author Response
The paper presents current data on the connection between Bell's palsy and Covid-19. This thorough research on the neurological manifestation of Covid-19, based on up-to-date literature, meets the requirements of the Journal and adds information to the general knowledge of Covid-19.
Reply: Thank you for the review and comments.
Reviewer 4 Report
This is an inclusive review of interesting literature on an issue common to many vaccines that may be of greater interest in the case of the COVID-19 vaccine.
The manuscript is generally well written and explained, however some forms of English need to be made clearer and more formal (e.g. In order to make conclusions about the two, there must 99 be an accurate background incidence of BP as well as accurately estimated rates of BP 100 among both the vaccinated and unvaccinated)
In my opinion, the point of view that attracts is undoubtedly the fact that it is plausible that SARSA-Cov2 has relations with the central nervous system (thinking that commonly coronaviruses can lead to neurogenic forms and that the onset of COVID-19 is often anosmia which can be understood as a neurological symptom) therefore it would be worth within the discussion to dwell on the relations with the central nervous system and nerves by recalling and evaluating papers that have already lent the focus to this topic (Armocida D, Palmieri M, Frati A, Santoro A, Pesce A. How SARS-Cov-2 can involve the central nervous system. A systematic analysis of literature from the department of human neurosciences of Sapienza University, Italy. J Clin Neurosci. 2020 Sep;79:231-236. doi: 10.1016/j.jocn.2020.07.007. Epub 2020 Jul 7. PMID: 33070902; PMCID: PMC7340069.)
Finally, the conclusions appear to be uncomprehensive and hasty.
Author Response
This is an inclusive review of interesting literature on an issue common to many vaccines that may be of greater interest in the case of the COVID-19 vaccine.
The manuscript is generally well written and explained, however some forms of English need to be made clearer and more formal (e.g. In order to make conclusions about the two, there must 99 be an accurate background incidence of BP as well as accurately estimated rates of BP 100 among both the vaccinated and unvaccinated)
Reply: Thank you, we have revised this sentence and others which may be unclear.
In my opinion, the point of view that attracts is undoubtedly the fact that it is plausible that SARSA-Cov2 has relations with the central nervous system (thinking that commonly coronaviruses can lead to neurogenic forms and that the onset of COVID-19 is often anosmia which can be understood as a neurological symptom) therefore it would be worth within the discussion to dwell on the relations with the central nervous system and nerves by recalling and evaluating papers that have already lent the focus to this topic (Armocida D, Palmieri M, Frati A, Santoro A, Pesce A. How SARS-Cov-2 can involve the central nervous system. A systematic analysis of literature from the department of human neurosciences of Sapienza University, Italy. J Clin Neurosci. 2020 Sep;79:231-236. doi: 10.1016/j.jocn.2020.07.007. Epub 2020 Jul 7. PMID: 33070902; PMCID: PMC7340069.)
Reply: Thank you, we have added this reference to the paper.
Finally, the conclusions appear to be uncomprehensive and hasty.
Reply: The conclusions have been edited and we hope that this is more comprehensive and clear.
Round 2
Reviewer 1 Report
The manuscript was improved. I recommend to accept in this version.
Reviewer 4 Report
Corrections accepted. Is suitable for publication